# Minimal-Entropy Correlation Alignment for Unsupervised Deep Domain Adaptation

**Pietro Morerio[1], Jacopo Cavazza[1] & Vittorio Murino[1,2]**
[1] Pattern Analysis and Computer Vision (PAVIS), Istituto Italiano di Tecnologia - Genova, Italy
[2] University of Verona, Department of Computer Science - Verona, Italy
`{pietro.morerio,jacopo.cavazza,vittorio.murino}@iit.it`

## Abstract

In this work, we face the problem of unsupervised domain adaptation with a novel deep learning approach which leverages our finding that entropy minimization is induced by the optimal alignment of second order statistics between source and target domains. We formally demonstrate this hypothesis and, aiming at achieving an optimal alignment in practical cases, we adopt a more principled strategy which, differently from the current Euclidean approaches, deploys alignment along geodesics. Our pipeline can be implemented by adding to the standard classification loss (on the labeled source domain), a source-to-target regularizer that is weighted in an unsupervised and data-driven fashion. We provide extensive experiments to assess the superiority of our framework on standard domain and modality adaptation benchmarks.

## 1 Introduction

Learning visual representations that are invariant across different domains is an important task in computer vision. Actually, data labeling is onerous and even impossible in some cases. It is thus desirable to train a model with full supervision on a *source*, labeled domain and then learn how to transfer it on a *target* domain, as opposed to retrain it completely from scratch. Moreover, the latter stage is actually not possible if the target domain is totally unlabelled: this is the setting we consider in our work. In the literature, this problem is known as *unsupervised domain adaptation* which can be regarded as a special *semi-supervised* learning problem, where labeled and unlabeled data come from different domains. Since no labels are available in the target domain, source-to-target adaptation must be carried out in a fully unsupervised manner. Clearly, this is an arguably difficult task since transferring a model across domains is complicated by the so-called *domain shift* [Torralba & Efros (2011)]. In fact, while switching from the source to the target, even if dealing with the same $K$ visual categories in both domains, different biases may arise related to several factors. For instance, dissimilar points of view, illumination changes, background clutter, etc.

In the previous years, a broad class of approaches has leveraged on *entropy optimization* as a proxy for (unsupervised) domain adaptation, borrowing this idea from semi-supervised learning [Grandvalet & Bengio (2004)]. By either performing entropy regularization [Tzeng et al. (2015); Carlucci et al. (2017); Saito et al. (2017)], explicit entropy minimization [Haeusser et al. (2017)], or implicit entropy maximization through adversarial training [Ganin & Lempitsky (2015); Tzeng et al. (2017)], this statistical tool has demonstrated to be powerful for adaptation purposes.

Alternatively, there exist methods which try to align the source to the target domain by learning an explicit transformation between the two so that the target data distribution can be matched to the one of the source one [Glorot et al. (2011); Kan et al. (2015); Shekhar et al. (2013); Gopalan & Li (2011); Gong et al. (2012a)]. Within this paradigm, *correlation alignment* minimizes the distance between second order statistics computed in the form of covariance representations between features from the source a [Fernando et al. (2013); Sun et al. (2016); Sun & Saenko (2016)].

Apparently, correlation alignment and entropy minimization may seem two unrelated and approaches in optimizing models for domain adaptation. However, in this paper, we will show that this is not the case and, indeed, we claim that the two classes of approaches are deeply intertwined. In addition to

formally discuss the latter aspect, we also obtain a solution for the prickly problem of hyperparameter validation in unsupervised domain adaptation. Indeed, one can construct a validation set out of source data but the latter is not helpful since not representative of target data. At the same time, due to the lack of annotations on the target domain, usual (supervised) validation techniques can not be applied.

In summary, this paper brings the following contributions.

1. We explore the two paradigms of correlation alignment and entropy minimization, by formally demonstrating that, at its optimum, correlation alignment attains the minimum of the sum of cross-entropy on the source domain and of the entropy on the target.

2. Motivated by the urgency of penalizing correlation misalignments in practical terms, we observe that an Euclidean penalty, as adopted in [Sun et al. (2016); Sun & Saenko (2016)], is not taking into account the structure of the manifold where covariance matrices lie in. Hence, we propose a different loss function that is inspired by a geodesic distance that takes into account the manifold's curvature while computing distances.

3. When aligning second order statistics, a hyper-parameter controls the balance between the reduction of the domain shift and the supervised classification on the source domain. In this respect, a manual cross-validation of the parameter is not straightforward: doing it on the source domain may not be representative, and it is not possible to do on the target due to the lack of annotations. Owing to our principled connection between correlation alignment and entropy regularization, we devise an entropy-based criterion to accomplish such validation in a data-driven fashion.

4. We combine the geodesic correlation alignment with the entropy-based criterion in a unique pipeline that we call *minimal-entropy correlation alignment*. Through an extensive experimental analysis on publicly available benchmarks for transfer object categorization, we certify the effectiveness of the proposed approach in terms of systematic improvements over former alignment methods and state-of-the-art techniques for unsupervised domain adaptation in general.

The rest of the paper is outlined as follows. In Section 2, we report the most relevant related work as background material. Section 3 presents our theoretical analysis which inspires our proposed method for domain adaptation (Section 4). We report a broad experimental validation in Section 5. Finally, Section 6 draws conclusions.

## 2 BACKGROUND AND RELATED WORK

In this Section, we will detail the two classes of correlation alignment and entropy optimization methods that are combined by our adaptation technique. An additional literature review is available in Appendix A.

We consider the problem of classifying an image $\mathbf{x}$ in a $K$-classes problem. To do so, we exploit a bunch of labeled images $\mathbf{x}_1, \ldots, \mathbf{x}_n$ and we seek for training a statistical classifier that, during inference, provides probabilities for a given test image $\bar{\mathbf{x}}$ to belong to each of the $K$ classes. In this work, such classifier is fixed to be a deep multi-layer feed-forward neural network denoted as

$$f(\bar{\mathbf{x}}; \theta) = [\mathbb{P}(\text{class}(\bar{\mathbf{x}}) = 1), \quad \mathbb{P}(\text{class}(\bar{\mathbf{x}}) = 2), \quad \ldots, \quad \mathbb{P}(\text{class}(\bar{\mathbf{x}}) = K)]. \quad (1)$$

The network $f$ depends upon some parameters/weights $\theta$ that are optimized by minimizing over $\theta$ the cross-entropy loss function

$$H(\mathbf{X}, \mathbf{Z}) = -\sum_{i=1}^{n} \langle \mathbf{z}_i, \log f(\mathbf{x}_i; \theta) \rangle. \quad (2)$$

In (2), for each image $\mathbf{x}_i$, the inner product $\langle \cdot, \cdot \rangle$ computes a similarity measure between the network prediction $f(\mathbf{x}_i; \theta)$ and the corresponding data label $\mathbf{z}_i$, which is a $K$ dimensional one-hot encoding vector. Precisely, $z_{ik} = 1$ if $\mathbf{x}_i$ belongs to the $k$-th class, being zero otherwise. Finally, for notational simplicity, let $\mathbf{X}$ and $\mathbf{Z}$ define the collection all images $\mathbf{x}_i$ and corresponding labels $\mathbf{z}_i$, respectively.

In a classical fully supervised setting, other than minimizing (2), one can also add some weighted additive regularizers to the final loss, such as an $L^2$ penalty. But, in the case of domain adaptation, $\theta$ should be chosen as to promote a good portability from the source $\mathcal{S}$ to the target domain $\mathcal{T}$.

*Correlation alignment.* In the case of unsupervised domain adaptation, we assume that none of the examples in the target domain is labelled and, therefore, we should perform adaptation at the feature level. In the case of correlation alignment, we can replace (2) with the following problem

$$\min_\theta \left[ H(\mathbf{X}_\mathcal{S}, \mathbf{Z}_\mathcal{S}) + \lambda \cdot \ell(\mathbf{C}_\mathcal{S}, \mathbf{C}_\mathcal{T}) \right], \qquad \lambda > 0, \tag{3}$$

where we compute the supervised cross-entropy loss between data $\mathbf{X}_\mathcal{S}$ and annotations $\mathbf{Z}_\mathcal{S}$ belonging to the source domain only. Concurrently, the network parameters $\theta$ are modified in order to align the covariance representations

$$\mathbf{C}_\mathcal{S} = \mathbf{A}_\mathcal{S}\mathbf{J}\mathbf{A}_\mathcal{S}^\top, \quad \text{and} \quad \mathbf{C}_\mathcal{T} = \mathbf{A}_\mathcal{T}\mathbf{J}\mathbf{A}_\mathcal{T}^\top \tag{4}$$

that are computed through the centering matrix $\mathbf{J}$ (see [Ha Quang et al. (2014); Cavazza et al. (2016)] for a closed-form) on top of the activations computed at a given layer[1] by the network $f(\cdot, \theta)$. Precisely, $\mathbf{A}_\mathcal{S}$ and $\mathbf{A}_\mathcal{T}$ stack by columns the $d$-dimensional activations computed from the source and the target domains. Also, $\theta$ is regularized according to the following Euclidean penalization

$$\ell(\mathbf{C}_\mathcal{S}, \mathbf{C}_\mathcal{T}) = \frac{1}{4d^2} \|\mathbf{C}_\mathcal{S} - \mathbf{C}_\mathcal{T}\|_F^2 \tag{5}$$

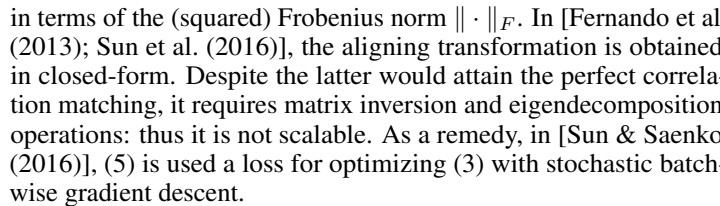

in terms of the (squared) Frobenius norm $\|\cdot\|_F$. In [Fernando et al. (2013); Sun et al. (2016)], the aligning transformation is obtained in closed-form. Despite the latter would attain the perfect correlation matching, it requires matrix inversion and eigendecomposition operations: thus it is not scalable. As a remedy, in [Sun & Saenko (2016)], (5) is used a loss for optimizing (3) with stochastic batch-wise gradient descent.

Figure 1: Geodesic versus Euclidean distances in the case of a non-zero curvature manifold (as the one of SPD matrices).

**Problem 1.** *Mathematically, covariance representations* (4) *are symmetric and positive definite (SPD) matrices belonging to a Riemannian manifold with non-zero curvature [Arsigny et al. (2007)]. Therefore, measuring correlation (mis)alignments with an Euclidean metric like* (5) *is arguably suboptimal since it does not capture the inner geometry of the data (see Figure 1).*

*Entropy regularization.* The cross entropy $H$ on the source domain and entropy $E$ on the target domain can be optimized as follows:

$$\min_\theta \left[ H(\mathbf{X}_\mathcal{S}, \mathbf{Z}_\mathcal{S}) + \gamma E(\mathbf{X}_\mathcal{T}) \right], \quad \gamma > 0 \tag{6}$$

where

$$E(\mathbf{X}_\mathcal{T}) = -\sum_{\mathbf{x}_t \in \mathcal{T}} \langle f(\mathbf{x}_t; \theta), \log f(\mathbf{x}_t; \theta) \rangle. \tag{7}$$

In this way, we circumvent the impossibility of optimizing the cross entropy on the target (due to the unavailability of labels on $\mathcal{T}$), and we replace it with the entropy $E(\mathbf{X}_\mathcal{T})$ computed on the soft-labels $\mathbf{z}_{\text{soft}}(\mathbf{x}_t) = f(\mathbf{x}_t; \theta)$, which is nothing but the network predictions [Lee (2013)]. Empirically, soft-labels increases the confidence of the model related to its prediction. However, for the purpose of domain adaptation, optimizing (6) is *not enough* and, in parallel, ancillary adaptation techniques are invoked. Specifically, either additional supervision [Tzeng et al. (2015)], batch normalization [Carlucci et al. (2017)] or probabilistic walk on the data manifold [Haeusser et al. (2017)] have been exploited. As a different setup, a min-max problem can be devised where $H(\mathbf{X}_\mathcal{S}, \mathbf{Z}_\mathcal{S})$ is minimized and, at the same time, entropy is maximized within a binary classification of predicting whether a given instance belongs to the source or the target domain. This is done in [Ganin & Lempitsky (2015)] and [Tzeng et al. (2017)] by reversing the gradients and using adversarial training, respectively. In practical terms, this means that, in addition to the loss function in (6), one needs to carry out other parallel optimizations whose reciprocal balance in influencing the parameters' update is controlled by means of hyper-parameters. Since the latter have to be grid-searched, a validation set is needed in order to select the hyper-parameters' configuration that corresponds to the best performance on it. How to select the aforementioned validation set leads to the following point.

---

[1]In principle, correlation alignment can be done at multiple layers in parallel, but empirical evidences [Sun et al. (2016); Sun & Saenko (2016)] suggest that a solid performance is achieved even if it's done only once.

**Problem 2.** *In the case of domain adaptation, cross-validation for hyper-parameter tuning on the source directly is unreasonable because of the domain shift. In fact, for instance, [Tzeng et al. (2015)] can do it only by adding supervision on the target and, in [Carlucci et al. (2017)], cross-validation is performed on the source* after *the target has been aligned to it. Since we need $\lambda$ to be fixed before solving for correlation alignment and since we consider a fully unsupervised adaptation setting, we cannot use any of the previous strategy and, obviously, we are not allowed for supervised cross-validation on the target. Thus, hyper-parameter tuning is really a problem.*

In this work, we combine the two classes of correlation alignment [Sun et al. (2016); Fernando et al. (2013); Sun & Saenko (2016)] and entropy optimization [Tzeng et al. (2015); Ganin & Lempitsky (2015); Tzeng et al. (2017); Haeusser et al. (2017); Carlucci et al. (2017)] in a unique framework. By doing so, we embrace a more principled approach to align covariance representations (as to tackle Problem 1), while, at the same time, solving Problem 2 with a novel unsupervised and data-driven cross-validation technique.

## 3   MINIMAL-ENTROPY CORRELATION ALIGNMENT

### 3.1   CONNECTIONS BETWEEN CORRELATION ALIGNMENT AND ENTROPY MINIMIZATION

In this section, we deploy a rigorous mathematical connection between correlation alignment and entropy minimization in order to understand the mutual relationships. The following theorem (see proof in Appendix B) represents the main result.

**Theorem 1.** *With the notation introduced so far, if $\theta^\star$ optimally aligns correlation in (3), then, $\theta^\star$ minimizes (6) for every $\gamma > 0$.*

The previous statement certifies that, at its optimum, correlation alignment provides minimal entropy for free. If one compares (3) with (6), one may notice that, in both cases, we are minimizing $H$ over the source domain $\mathcal{S}$. Therefore, if we assume that $H(\mathbf{X}_{\mathcal{S}}, \mathbf{Z}_{\mathcal{S}}) = \min$, we have a perfect classifier whose predictions on $\mathcal{S}$ are extremely confident and correct. Thus, the predictions are distributed in a very picky manner and, therefore, entropy on the source is minimized. At the same time, we can minimize the entropy on the target since $\mathcal{T}$ is made "indistinguishable" from $\mathcal{S}$ after the alignment. Hence, the target's predictions are distributed in a similar picky way so that entropy on $\mathcal{T}$ is minimized as well.

**Observation 1.** *Since we proved that optimal correlation alignment implies entropy minimization, one may ask whether the converse holds. That is, if the optimum of (6) gives the optimum of (3). The answer is negative as it will be clear by the following counterexample. In fact, we can always minimize the cross entropy on the source with a fully supervised training on $\mathcal{S}$. However, such classifier could be always confident in classifying a target example as belonging to, say, Class 1. After that, we can deploy a dummy adaptation step that, for whatever target image $\bar{\mathbf{x}}$ to be classified, we always predict it to be Class 1. In this case the entropy on the target is clearly minimized since the distribution of the target prediction is a Dirac's delta $\delta_{1k}$ for any class $k$. But, obviously, nothing has been done for the sake of adaptation and, in particular, optimal correlation alignment is far from being realized (see Appendix C).*

In Theorem 1, the assumption of having an optimal correlation alignment is crucial for our theoretical analysis. However, in practical terms, optimal alignment is also desirable in order to effectively deploy domain adaptation systems. Moreover, despite the optimal alignment in (3) is able to minimize (6) for any $\gamma > 0$, in practice, hyper-parameters need to be cross-validated and this is not an easy task in unsupervised domain adaptation (as we explained in Problem 2). In the next section, a solution for all these problems will be distilled from our improved knowledge.

## 4   UNSUPERVISED DEEP DOMAIN ADAPTATION BY MINIMAL-ENTROPY CORRELATION ALIGNMENT

Based on the previous remarks, we address the unsupervised domain adaptation problem by training a deep net for supervised classification on $\mathcal{S}$ while adding a loss term based on a geodesic distance

on the SPD manifold. Precisely, we consider the (squared) log-Euclidean distance

$$\ell_{log}(\mathbf{C}_\mathcal{S}, \mathbf{C}_\mathcal{T}) = \frac{1}{4d^2}\big\|\mathbf{U}\mathrm{diag}(\log(\sigma_1), \dots, \log(\sigma_d))\mathbf{U}^\top - \mathbf{V}\mathrm{diag}(\log(\mu_1), \dots, \log(\mu_d))\mathbf{V}^\top\big\|_F^2 \quad (8)$$

where $d$ is the dimension of the activations $\mathbf{A}_\mathcal{S}$ and $\mathbf{A}_\mathcal{T}$, whose covariances are intended to be aligned, $\mathbf{U}$ and $\mathbf{V}$ are the matrices which diagonalize $\mathbf{C}_\mathcal{S}$ and $\mathbf{C}_\mathcal{T}$, respectively, and $\sigma_i$, $\mu_i$, $i = 1, \dots, d$ are the corresponding eigenvalues. The normalization term $1/d^2$ accounts for the sum of the $d^2$ terms in the $\|\cdot\|_F^2$ norm, which makes $\ell_{log}$ independent from the size of the feature layer.

The geodesic alignment for correlation is attained by minimizing the problem $\min_\theta\left[H(\mathbf{X}_\mathcal{S}, \mathbf{Z}_\mathcal{S}) + \lambda \cdot \ell_{log}(\mathbf{C}_\mathcal{S}, \mathbf{C}_\mathcal{T})\right]$, for some $\lambda > 0$. This allows lo learn good features for classification which, at the same time, do not overfit the source data since they reflect the statistical structure of the target set. To this end, a geodesic distance accounts for the geometrical structure of covariance matrices better than (3). In this respect, the following two aspects are crucial.

- *Problem 1* is addressed by introducing the log-Euclidean distance $\ell_{log}$ between SPD matrices, which is a geodesic distance widely adopted in computer vision [Cavazza et al. (2016); Zhang et al. (2016); Ha Quang et al. (2014; 2016); Cavazza et al. (2017)] when dealing with covariance operators. The rationale is that, within the many geodesic distances, (8) is extremely efficient because does not require matrix inversions (like the affine one $\ell_{\mathrm{aff}}(\mathbf{C}_\mathcal{S}, \mathbf{C}_\mathcal{T}) = \|\log(\mathbf{C}_\mathcal{S}\mathbf{C}_\mathcal{T}^{-1})\|_F$). Moreover, while shifting from one geodesic distance to another, the gap in performance obtained are negligible, provided the soundness of the metric [Zhang et al. (2016)].

- As observed in *Problem 2*, the hyperparameter $\lambda$ is a critical coefficient to be cross validated. In fact, a high value of $\lambda$ is likely to force the network towards learning oversimplified low-rank feature representations. Despite this may result in perfectly aligned covariances, it could be useless for classification purposes. On the other hand, a small $\lambda$ may not be enough to bridge the domain shift. Motivated by Theorem 1, we select the $\lambda$ which minimizes the entropy $E(\mathbf{X}_\mathcal{T})$ on the target domain. Indeed, since we proved that $H(\mathbf{X}_\mathcal{S})$ is minimized at the same time in both (3) and (6), we can naturally tune $\lambda$ so that $E(\mathbf{X}_\mathcal{T}) = \min$. Note that this entropy-based criterion for $\lambda$ is totally fair in unsupervised domain adaptation since, as in (6), $E$ does not require ground truth target labels to be computed, but only relies on inferred soft-labels.

In summary, we propose the following minimization pipeline for unsupervised domain adaptation, which we name *Minimal-Entropy Correlation Alignment* (*MECA*)

$$\min_\theta\left[H(\mathbf{X}_\mathcal{S}, \mathbf{Z}_\mathcal{S}) + \lambda \cdot \ell_{log}(\mathbf{C}_\mathcal{S}, \mathbf{C}_\mathcal{T})\right] \qquad \text{subject to} \qquad \lambda \text{ minimizes } E(\mathbf{X}_\mathcal{T}). \quad (9)$$

In other words, in (9), we minimize the objective functional $H(\mathbf{X}_\mathcal{S}, \mathbf{Z}_\mathcal{S}) + \lambda \cdot \ell_{log}(\mathbf{C}_\mathcal{S}, \mathbf{C}_\mathcal{T})$ by gradient descent over $\theta$. While doing so, we can choose $\lambda$ by validation, such that the network $f(\cdot; \theta)$ is able, at the same time, to attain the minimal entropy on the target domain.

*Differentiability.* For a fixed $\lambda$, the loss (9) needs to be differentiable in order for the minimization problem to be solved via back-propagation, and its gradients should be calculated with respect to the input features. However, as (4) shows, $\mathbf{C}_\mathcal{S}$ and $\mathbf{C}_\mathcal{T}$ are polynomial functions of the activations and the same holds when one applies the Euclidean norm $\|\cdot\|_F^2$. Additionally, since the $\log$ function is differentiable over its domain, we can easily see that we can still write down the gradients of the loss (9) in a closed form by exhaustively applying the chain rule over elementary functions that are in turn differentiable. In practice, this is not even needed, since modern tools for deep learning consist in software libraries for numerical computation whose core abstraction is represented by *computational graphs*. Single mathematical operations (e.g., matrix multiplication, summation etc.) are deployed on nodes of a graph, and data flows through edges. Reverse-mode differentiation takes advantage of the gradients of single operations, allowing training by backpropagation through the graph. The loss (9) can be easily written (for a fixed $\lambda$) in few lines of code by exploiting mathematical operations which are already implemented, together with their gradients, in TensorFlow™ or other libraries[2].

## 5 RESULTS

In this Section we will corroborate our theoretical analysis with a broad validation which certify the correctness of Theorem 1 and the effectiveness of our proposed entropy-based cross-validation

---

[2]Code available at https://github.com/pmorerio/minimal-entropy-correlation-alignment

for $\lambda$ in (9). In addition, by means of a benchmark comparison with state-of-the-art approaches in unsupervised domain adaptation, we will prove the effectiveness of the geodesic versus the Euclidean alignment and, in general, that MECA outperforms many previously proposed methods.

We run the following adaptation experiments. We use digits from SVHN [Netzer et al. (2011)] as source and we transfer on MNIST. Similarly, we transfer from SYN DIGITS [Ganin & Lempitsky (2015)] to SVHN. For the object recognition task, we train a model to classify objects on RGB images from NYUD [Silberman et al. (2012)] dataset and we test on (different) depth images from the same visual categories. Reproducibility details for both dataset and baselines are reported in Appendix D.

## 5.1 NUMERICAL EVIDENCES FOR OUR THEORETICAL ANALYSIS

As shown in Theorem 1, correlation alignment and entropy regularization are intertwined. Despite this result holds at the optimum only, we can actually observe an even stronger linkage. Precisely, we empirically register that a gradient descent path for correlation alignment induces a gradient descent path for entropy minimization. In fact, in the top-left part of Figure 2, while running correlation alignment to align source and target with either an Euclidean (red curve) or geodesic penalty (orange curve), we are able to minimize the entropy. Also, when comparing the two, geodesic provides a lower entropy value than the Euclidean alignment, meaning that our approach is able to better minimize $E(\mathbf{X}_\mathcal{T})$. Interestingly, even if the baseline with no adaptation is able to minimize the entropy as well (blue curve), this is only a matter of overfitting the source. In fact, the baseline produces a classifier which is overconfidently wrong on the target (as explained in Appendix C) as long as training evolves. Remember that optimal correlation alignment implies entropy minimization being the converse not true: if we check the alignment of source and target distributions (Figure 2 bottom-left), we see that, with no adaptation (blue curve), the two distributions are increasingly mismatched as long as training proceeds. Differently, with either Euclidean or geodesic alignments, we are able to match the two and, in order to check the quality of such alignment, we conduct the following experiment.

In Figure 2, right column, we show the plots of target entropy and classification accuracies related to SVHN→MNIST as a function of $\lambda \in \{0.1, 0.5, 1, 2, 5, 7, 10, 20\}$. Let us stress that, since we measure distances on the SPD manifold directly, we can conjecture that (8) can achieve a better alignment between covariances than (5). Actually, if one applies the closed-form solution of [Sun et al. (2016)] the optimal alignment can be found analytically. However, due to the required matrix inversions, such approach is not scalable an one needs to backpropagate errors starting from a penalty function in order to train the model. As one can clearly see in Figure 2 (right), Euclidean alignment is performing about 5% worse than our proposed geodesic alignment on SVHN→MNIST. But, most importantly, in the Euclidean case, the minimal entropy does not correspond to the maximum performance on the target. Differently, when using the geodesic penalty (8), we see that the $\lambda$ which minimizes $E(\mathbf{X}_\mathcal{T})$ is also the one that gives the maximum performance on the target. Thus, we can conclude that our geodesic approach is better than the Euclidean one since totally compatible with a data-driven cross-validation strategy for $\lambda$, requiring no labels belonging to the target domain.

Additional evidences of the superiority of our proposed geodesic alignment in favor of a classical one are reported in the next Section. Thereby, our Minimal-Entropy Correlation Alignment (*MECA*) method is benchmarked against state-of-the-art approaches for unsupervised deep domain adaptation.

## 5.2 COMPARATIVE EVALUATION OF *MECA*

In this Section, we benchmark MECA against general state-of-the-art frameworks for unsupervised domain adaptation with deep learning: Domain Separation Network (DSN) [Bousmalis et al. (2016)] and Domain Transfer Network (DTN) [Taigman et al. (2017)]. In addition, we also compare with two (implicit) entropy maximization frameworks - Gradient Reversal Layer (GRL) [Ganin & Lempitsky (2015)] and ADDA [Tzeng et al. (2017)] - and with the entropy regularization technique of [Saito et al. (2017)], which uses a *triple* classifier (TRIPLE). Also, we consider the deep Euclidean correlation alignment named Deep CORAL [Sun & Saenko (2016)]. In order to carry on a comparative analysis, we setup standard baseline architectures which reproduce *source only* performances (i.e., performance of the models with no adaptation). More details are provided in Appendix D

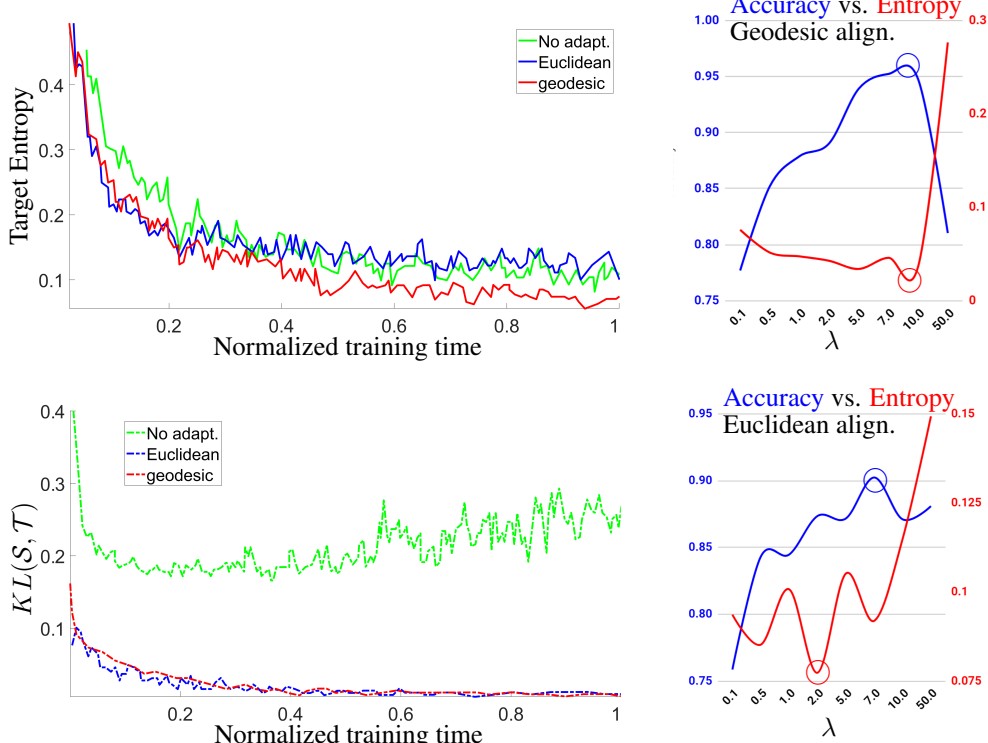

Figure 2: A gradient descent path for correlation alignment induces a gradient descent path for entropy minimization. *Left column.* We compare a baseline CNN trained on the source (SVHN) only (blue), with the same model where we applied either Euclidean (red) or geodesic alignment (orange) with $\lambda = 0.1$ using MNIST as target. We compare the target entropy (top) and the correlation alignment (bottom) with a $KL$ divergence between source and target distribution. *Right column.* Target accuracy versus target entropy as a function of $\lambda$ for Euclidean (bottom) or geodesic (top) correlation alignment. Best viewed in colors.

In all cases, we report the published results from the other competitors, even when they devised more favorable experimental conditions than ours (*e.g.*, DTN exploits the extra data provided with SVHN). In the case of Deep CORAL, since the published results only cover the (almost saturated) Office dataset, we decided to run our own implementation of the method. While doing this, in order to cross-validate $\lambda$ in (3), we tried to balance the magnitudes of the two losses (2) and (5) as prescribed in the original work. However, since this approach does not provide good results, we were forced to cross-validate Deep Coral on the target directly. Let us remark that, as we show in Section 5.1, our proposed entropy-based cross validation is not always compatible with an Euclidean alignment. Differently, for MECA, our geodesic approach naturally embeds the entropy-based criterion and, consequently, we are able to maximize the performance on the target with a fully unsupervised and data-dependent cross-validation.

In addition, the classification performance registered by MECA is extremely solid. In fact, in the worst case we found (SYN→SVHN), MECA is performing practically on par with respect to Deep CORAL, despite for the latter labels on the target are used, being not far from the score of TRIPLE. This point can be explained with the fact that, for some benchmark datasets, the domain shift is not so prominent - e.g., check the visual similarities between SYN and SVHN datasets in the first two columns of Figure 3. In such cases, one can naturally argue that the type of alignment is not so crucial since adaptation is not strictly necessary, and the two types of alignment are pretty equivalent. This also explains the gap shown by MECA from the state-of-the-art (TRIPLE, 93.1%, which performs better than training on target with our architecture) and, eventually, the fact that the baseline itself is already doing pretty well (87.0%). As the results certify, MECA is systematically outperforming Deep CORAL: +0.5% on SYN→SVHN, +2.1% on NYUD and +5% on SVHN→MNIST.

Table 1: Unsupervised domain adaptation with *MECA*. Perfomance is measured as normalized accuracy and we compare with general, entropy-related (E) and correlation alignment (C) state-of-the-art approaches. §We also include this experiment exclusively for evaluation purposes. Let us stress that all methods in comparisons and our proposed MECA exploit labels only from the source domain during training. †A more powerful feature extractor as baseline and uses also extra SVHN data. ‡Results refer to our own Tensorflow[TM] implementation, with cross-validation on the target.

| Method | | SVHN→MNIST | NYUD | SYN→SVHN |
|---|---|---|---|---|
| *Source only: baseline* | | *0.685* | *0.139* | *0.870* |
| *Train on target*§ | | *0.994* | *0.468* | *0.922* |
| DSN [Bousmalis et al. (2016)] | | 0.827 | - | 0.912 |
| DTN† [Taigman et al. (2017)] | | 0.844 | - | - |
| GRL [Ganin & Lempitsky (2015)] | (E) | 0.739 | - | 0.911 |
| ADDA [Tzeng et al. (2017)] | (E) | 0.760 | 0.211 | - |
| TRIPLE [Saito et al. (2017)] | (E) | 0.862 | - | **0.931** |
| Deep CORAL‡ [Sun & Saenko (2016)] | (**C**) | 0.902 | 0.224 | 0.898 |
| *MECA* (proposed) | (**E + C**) | ***0.952*** | ***0.255*** | *0.903* |

Finally, our proposed MECA is able to improve the previous methods by margin on SVHN→MNIST (+5.0%) and on NYUD as well (+2.6%).

## 6 CONCLUSIONS

In this paper we carried out a principled connection between correlation alignment and entropy minimization, formally demonstrating that the optimal solution to the former problem gives for free the optimal solution of the latter. This improved knowledge brought us to two algorithmic advances. First, we achieved a more effective alignment of covariance operators which guarantees a superior performance. Second, we derived a novel cross-validation approach for the hyper-parameter $\lambda$ so that we can obtain the maximum performance on the target, even not having access to its labels. These two components, when combined in our proposed MECA pipeline, provide a solid performance against state-of-the-art methods for unsupervised domain adaptation.

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

# APPENDICES

## A  A REVIEW OF CORRELATION ALIGNMENT AND ENTROPY OPTIMIZATION METHODS FOR DOMAIN ADAPTATION

For the problem of (unsupervised) domain adaptation, a first class of methods aims at learning transformations which align feature representations in the source and target sets. For instance, in [Glorot et al. (2011)] auto-encoders are exploited to learn common features. In [Kan et al. (2015)], a bi-shifting auto-encoder (BSA) is instead intended to shift source domain samples into target ones and, similarly, other methods approach the same problem by means of techniques based on dictionary learning (as in [Shekhar et al. (2013)]). Geodesic methods (such as [Gopalan & Li (2011); Gong et al. (2012a)] aim at projecting source and target datasets on a common manifold in such a way that the projection already solves the alignment problem. The approaches [Gong et al. (2012b); Gopalan et al. (2011)] learns a smooth transition between the source and data manifold by means of Principal Components Analysis and Partial Least Squares, respectively. Inspired by the idea of adapting second order statistics between the two domains, [Sun et al. (2016); Fernando et al. (2013)] propose a transformation to minimize the distance between the covariances of source and target datasets in order to, ultimately, achieve *correlation alignment*. Due to the well known properties of covariance operators, in some cases [Sun et al. (2016)], the alignment can be written down in closed-form. But, since the latter operation can be prohibitively expensive in terms of computational cost, Sun & Saenko (2016) implements correlation alignment in an end-to-end fashion by means of backpropagation.

A complementary family of approaches exploit the powerful statistical tool of *entropy optimization* in order to carry out adaptation. Indeed, the notion of association [Haeusser et al. (2017)] is actually implementing explicit entropy minimization [Grandvalet & Bengio (2004)] to align the target to the source embedding by navigating the data manifold by means of closed cyclic paths that interconnect instances belonging to the same objects' classes.
In parallel, there are cases [Ganin & Lempitsky (2015); Tzeng et al. (2017)] where minimax optimization is responsible for doing the following adversarial training. One seeks for feature representations that are effective for the primary visual recognition task being at the same time invariant while changing from source to target. The latter stage is implemented as the attempt of devising a random chance classifier which is asked to detect whether a given feature vector has been computed from a source or target data instance. Therefore, those approaches are implicitly promoting *entropy maximization*[3] at the classifier level.
Finally, entropy regularization is accomplished in [Tzeng et al. (2015); Carlucci et al. (2017); Saito et al. (2017)] as a complementary step to boost adaptation. Indeed, already established techniques for adaptation such as Batch Normalization [Ioffe & Szegedy (2015); Li et al. (2016)] are applied in low-level layers to align the representations. On top of that, adaptation is refined at the end of the feature hierarchy by introducing a entropy-based regularizer on the target domain based. Practically, the latter exploits network's prediction to generate pseudo-labels [Lee (2013); Tzeng et al. (2015); Carlucci et al. (2017); Saito et al. (2017)] and compensate for the lack of annotations on the target.

## B  PROOF OF THEOREM 1

*Proof.* By hypothesis, we assume that $\theta^\star$ is the optimal hyper-parameter which attains the optimum of (3), which implies

$$H(\mathbf{X}_\mathcal{S}, \mathbf{Z}_\mathcal{S}) = \min \quad \text{and} \quad \mathbf{C}_\mathcal{S} = \mathbf{C}_\mathcal{T}, \tag{10}$$

by the properties of the squared-distance function $d$.

Let us fix an arbitrary $\gamma > 0$ and let us consider

$$L(\theta) = H(\mathbf{X}_\mathcal{S}, \mathbf{Z}_\mathcal{S}) + \gamma E(\mathbf{X}_\mathcal{T}). \tag{11}$$

---

[3]Remember that the distribution that maximes the entropy is the uniform one and, clearly, the latter is the distribution that represents the prediction accomplished by a random chance classifier

the objective functional in (6) which rewrites

$$L(\theta) = -\sum_{\mathbf{x}_i \in \mathcal{S}} \log \left( \sum_{k=1}^{K} z_{ik} f_k(\mathbf{x}_i; \theta) \right) - \gamma \sum_{\mathbf{x}_j \in \mathcal{T}} \sum_{k=1}^{K} f_k(\mathbf{x}_j; \theta) \log \left( f_k(\mathbf{x}_j; \theta) \right) \qquad (12)$$

while writing down the expression of the cross-entropy function $H$ between ground truth source labels $Z_\mathcal{S}$ and network's predictions which are also exploited to compute the entropy function $E$ on the target domain.

By hypothesis, since $\theta^\star$ is such that $H(\mathbf{X}_\mathcal{S}, \mathbf{Z}_\mathcal{S}) = \min$, then the thesis will follow if we prove that

$$E(\mathbf{X}_\mathcal{T}) = -\gamma \sum_{\mathbf{x}_j \in \mathcal{T}} \sum_{k=1}^{K} f_k(\mathbf{x}_j; \theta^\star) \log \left( f_k(\mathbf{x}_j; \theta^\star) \right) = \min \qquad (13)$$

since the minimum of the sum of two functions is achieved when the two addends are minimized separately.

Now, by hypothesis, since we assume the optimal correlation alignment, then, due to the fact that $\mathbf{C}_\mathcal{S} = \mathbf{C}_\mathcal{T}$, we can assume that the statistical properties of the trained classifier on the source can be transferred to the target with null performance degradation since, basically, we have obtained the way to completely solve the domain shift issue. This implies that, if we assume that some oracle will provide us the ground truth labels $\mathbf{z}_j$ for the target domain, we can get that

$$f(\mathbf{x}_j; \theta^\star) = \mathbf{z}_j \qquad (14)$$

for any arbitrary $\mathbf{x}_j$ in the target domain $\mathcal{T}$. Note that $\theta^\star$ was optimized in a fair manner, by exploiting the labels of the source domain only and the fact that a perfect classification on the target is achieved is a side effect of assuming that we achieved the optimal correlation alignment, making the target data distribution essentially indistinguishable from the source one. In particular, $f(\mathbf{x}_j; \theta^\star)$ is a Dirac's delta function such that $f_k(\mathbf{x}_j; \theta^\star) = 1$ if $\mathbf{x}_j$ belongs to the $k$-th class and $f_k(\mathbf{x}_j; \theta^\star) = 0$ otherwise. Therefore, we get

$$-\gamma \sum_{\mathbf{x}_j \in \mathcal{T}} \sum_{k=1}^{K} f_k(\mathbf{x}_j; \theta^\star) \log \left( f_k(\mathbf{x}_j; \theta^\star) \right) = -\gamma \sum_{\mathbf{x}_j \in \mathcal{T}} \left[ \sum_{k \neq \text{class}(\mathbf{x}_j)} 0 + \log 1 \right] \qquad (15)$$

due to the fact that $_k(\mathbf{x}_j; \theta^\star)$ is a Dirac's delta and since we decompose, for each $\mathbf{x}_j$, the summation over $k$ in two parts: when $k$ equals the class of $\mathbf{x}_j$, $f_k(\mathbf{x}_j; \theta^\star) \log \left( f_k(\mathbf{x}_j; \theta^\star) \right) = \log 1 = 0$ and, in all other cases, the addends vanishes. Therefore

$$-\gamma \sum_{\mathbf{x}_j \in \mathcal{T}} \sum_{k=1}^{K} f_k(\mathbf{x}_j; \theta^\star) \log \left( f_k(\mathbf{x}_j; \theta^\star) \right) = 0. \qquad (16)$$

Since $E(\mathbf{X}_\mathcal{T})$ is a non-negative function, (16) gives the thesis (13) due to the generality of $\gamma$ □

## C  TARGET ENTROPY MINIMIZATION IS A NECESSARY, NOT SUFFICIENT CONDITION FOR DOMAIN ADAPTATION

Consider the fully supervised classification problem of optimizing $\theta$ for the deep neural network $f(\cdot, \theta)$ such that, while comparing network's prediction $f(\mathbf{x}_i, \theta)$ and ground truth annotations $\mathbf{z}_i$, *relative to the source domain* $\mathcal{S}$, we get the problem of

$$\text{training the network } f(\ \cdot\ ; \theta) \text{ such that } H(\mathbf{X}_\mathcal{S}, \mathbf{Z}_\mathcal{S}) = \min \qquad (17)$$

where, in (17), minimization is carried out on $\theta$. Now, we can devise a dummy classifier $\widetilde{f}$, depending upon the same exact parameter choice $\theta$ such that

$$\widetilde{f}(\bar{\mathbf{x}}; \theta) = \begin{cases} f(\bar{\mathbf{x}}; \theta) & \text{if } \bar{\mathbf{x}} \in \mathcal{S} \\ [1, 0, \ldots, 0] & \text{if } \bar{\mathbf{x}} \in \mathcal{T}. \end{cases} \qquad (18)$$

That is, we use on the target the same exact classifier that we trained on the source (with no adaptation). That is, source data is classified by $\widetilde{f}$ based on $f$, while, when asked to classify an image from the

target domain, $\widetilde{f}$ will *always* predict that instance to belong to the first class. By using the same exact scheme of proof as in Appendix B, we can show that, $\widetilde{f}$ achieves the minimal entropy $E(\mathbf{X}_{\mathcal{T}})$ on the target domain $\mathcal{T}$. This is an evidence for the fact that, although optimal correlation alignment implies minimal entropy, the converse is not true. Ancillary, it explains why in [Tzeng et al. (2015); Carlucci et al. (2017)], adaptation is effectively carried out with ancillary techniques and entropy regularization it's just a boosting factor as opposed to a factual regularizer for domain adaptation.

# D    TECHNICAL DETAILS

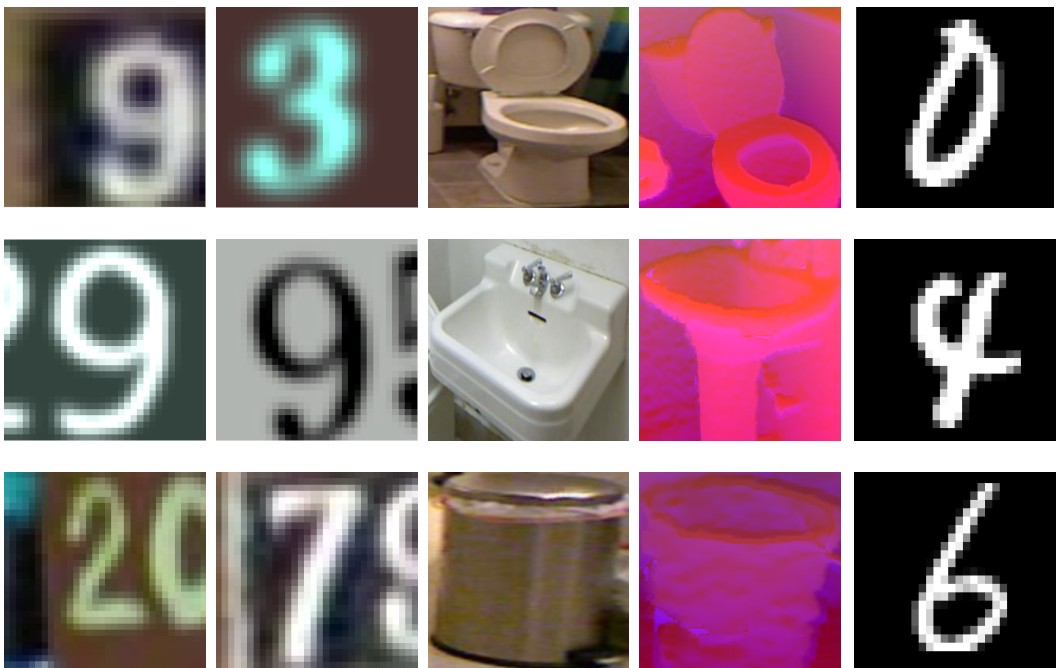

Figure 3: Sampled images from the datasets involved in the domain adaptation experiments. From left to right, SVHN (first column, digits 9, 9, 2 from top to bottom), SYN (second column, digits 3, 9, 7 from top to bottom), NYUD RGB (third column, `toilet`, `sink` and `garbage-bin` classes acquires as RGB), NYUD depth (fourth column, different instances from the same previous classes acquired with the alternative modality) and the well known MNIST dataset (fifth column).

## D.1    DATASETS

**SVHN $\rightarrow$ MNIST.** This split represents a very realistic domain shift, since SVHN [Netzer et al. (2011)] (Street-View-House-Numbers) is built with real-world house numbers. We used the whole training sets of both datasets, following the usual protocol for unsupervised domain adaptation (SVHN's training set contains $73,257$ images). We also resized MNIST images to $32 \times 32$ pixels and converted SVHN to grayscale, according to the standard protocol.

**NYUD (RGB $\rightarrow$ depth).** This domain adaptation problem is actually a *modality adaptation* task and it was recently proposed by Tzeng et al. [Tzeng et al. (2017)]. The dataset is gathered by cropping out object bounding boxes around instances of 19 classes of the NYUD [Silberman et al. (2012)] dataset. It comprises 2,186 labeled source (RGB) images and 2,401 unlabeled target depth images, HHA-encoded [Gupta et al. (2014)]. Note that these are obtained from two different splits of the original dataset, in order to ensure that the same instance is not seen in both domains. The adaptation task is extremely challenging, due to the very different nature of the data, the limited number of examples (especially for some classes) and the low resolution anf heterogeneous size of the cropped bounding boxes.

**SYN DIGITS → SVHN.** This split represents a synthetic-to-real domain adaptation problem, of great interest for research in computer vision, since often requires less efforts generating labeled synthetic data than obtaining large labeled dataset with real samples. SYN DIGITS [Ganin & Lempitsky (2015)] contains $500,000$ images belonging to the same SVHN's classes.

## D.2 Baseline architecture details

**SVHN → MNIST.**
The architecture is the very same employed in [Ganin & Lempitsky (2015)] with the only difference that the last fully connected layer (`fc2`) has only 64 units instead of 2048. Performances are the same, but covariance computation is less onerous. `fc2` is in fact the layer where domain adaptation i performed.

**NYUD (RGB → depth).**
We finetune a VGG in order to be comparable with ADDA baseline in [Tzeng et al. (2017)]. Covariance alignment occurs at `fc8`, which is replaced with a 64-unit layer.

**SYN DIGITS → SVHN.**
Same as for **SVHN → MNIST**, but `fc1` has 3072 units.

# E VISUALIZATIONS

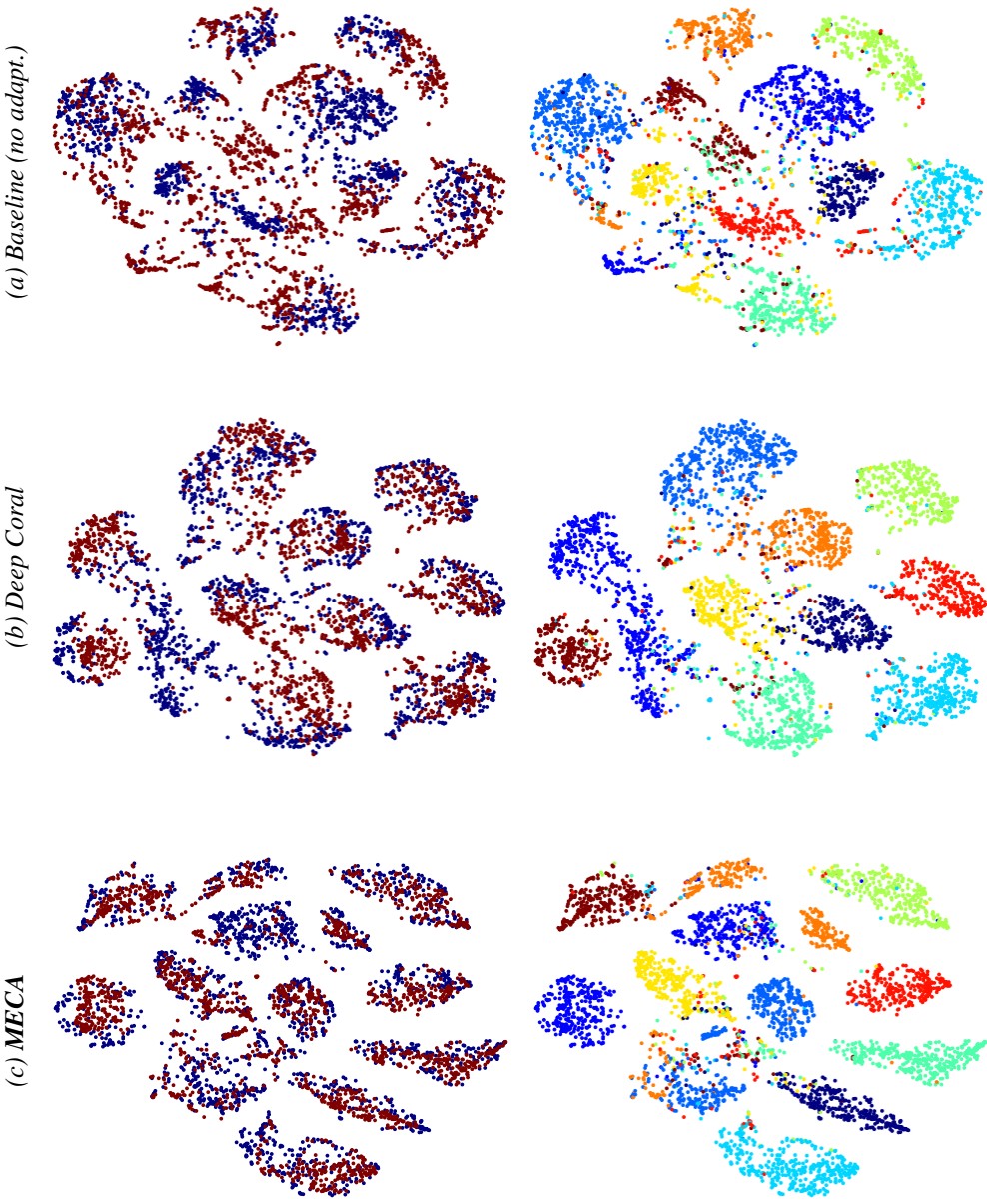

Figure 4: **SVHN → MNIST:** t-SNE [van der Maaten & Hinton (2008)] visualizations (64-dimensional features). *Left*: blue and red dots indicate SVHN (source) and MNIST (target) features, respectively. *Right*: different colors indicate the ten different classes. While *source* data (*blue*) is always well clustered, *target* data (*red*) is not. Correlation alignment in *(b)* and *(c)* makes the target distribution increasingly more similar to the source one. As qualitatively shown in the plots, *MECA* provides both better target clustering and domain similarity within the each cluster. This confirms the the quantitative results of Table 1, where *MECA* discriminates better than *Deep Coral*.

