# OpenReview forum: "Minimal-Entropy Correlation Alignment for Unsupervised Deep Domain Adaptation"
_ICLR.cc/2018/Conference — Accept (Poster)_

### Official Review · AnonReviewer1 · 2017-11-26
**Need further exploration for the use of entropy to select free parameters; geodesic correlation alignment is a reasonable improvement**

**Rating:** 6
**Confidence:** 5

**Review:**

This paper improves the correlation alignment approach to domain adaptation from two aspects. One is to replace the Euclidean distance by the geodesic Log-Euclidean distance between two covariance matrices. The other is to automatically select the balancing cost by the entropy on the target domain. Experiments are conducted from SVHN to MNIST and from SYN MNIST to SVHN. Additional experiments on cross-modality recognition are reported from RGB to depth.

Strengths:
+ It is a sensible idea to improve the Euclidean distance by the geodesic Log-Euclidean distance to better explore the manifold structure of the PSD matrices.
+ It is also interesting to choose the balancing cost using the entropy on the target. However, this point is worth further exploring (please see below for more detailed comments).
+ The experiments show that the geodesic correlation alignment outperforms the original alignment method.

Weaknesses:
- It is certainly interesting to have a scheme to automatically choose the hyper-parameters in unsupervised domain adaptation, and the entropy over the target seems like a reasonable choice. This point is worth further exploring for the following reasons.
1. The theoretical result is not convincing given it relies on many unrealistic assumptions, such as the null performance degradation under perfect correlation alignment, the Dirac’s delta function as the predictions over the target, etc.
2. The theorem actually does not favor the correlation alignment over the geodesic alignment. It does not explain that, in Figure 2, the entropy is able to find the best balancing cost \lamba for geodesic alignment but not for the Euclidean alignment.
3. The entropy alignment seems an interesting criterion to explore in general. Could it be used to find fairly good hyper-parameters for the other methods? Could it be used to determine the other hyper-parameters (e..g, learning rate, early stopping) for the geodesic alignment?
4. If one leaves a subset of the target domain out and use its labels for validation, how different would the selected balancing cost \lambda differ from that by the entropy?

- The cross-modality setup (from RGB to depth) is often not considered as domain adaptation. It would be better to replace it by another benchmark dataset. The Office-31 dataset is still a good benchmark to compare different methods and for the study in Section 5.1, though it is not necessary to reach state-of-the-art results on this dataset because, as the authors noted, it is almost saturated.

Question:
- I am not sure how the gradients were computed after the eigendecomposition in equation (8).


I like the idea of automatically choosing free parameters using the entropy over the target domain. However, instead of justifying this point by the theorem that relies on many assumptions, it is better to further test it using experiments (e.g., on Office31 and for other adaptation methods). The geodesic correlation alignment is a reasonable improvement over the Euclidean alignment.

---

> ### Author Response · Authors · 2017-12-28
> **Response to AnonReviewer1 - part 1**
>
> We thank the reviewer for having read our work with great detail and for the valuable suggestions. We will address all quoted weaknesses (W) and questions (Q) separately.
>
>
> W 1. - The theoretical result is not convincing given it relies on many unrealistic assumptions, such as the null performance degradation under perfect correlation alignment, the Dirac’s delta function as the predictions over the target, etc.
>
> In Theorem 1, by assuming the optimal correlation alignment, we can prove that entropy is minimized (which, ancillary, implies the Dirac’s delta function for the predictions). Under a theoretical standpoint, the strong assumption is balanced by the significant claim we have proved. In practical terms, the reviewers is right in observing that the optimal alignment is not granted for free, and this justifies the choice of a more sound metric for correlation alignment. That’s why we proposed the log-Euclidean distance to make the alignment closer to the optimal one.
> --
>
> W 2. - The theorem actually does not favor the correlation alignment over the geodesic alignment. It does not explain that, in Figure 2, the entropy is able to find the best balancing cost \lamba for geodesic alignment but not for the Euclidean alignment.
>
> As we showed in Figure 2, in the case of geodesic alignment, entropy minimization always correlate with the optimal performance on the target domain. Since the same does not always happen when an Euclidean metric is used, this is an evidence that Euclidean alignment is not able to achieve an optimal correlation alignment which, in comparison, is better achieved through our geodesic approach.
> --
>
> W 3. - The entropy alignment seems an interesting criterion to explore in general. Could it be used to find fairly good hyper-parameters for the other methods? Could it be used to determine the other hyper-parameters (e.g., learning rate, early stopping) for the geodesic alignment?
>
> It does make sense to fine tune the \lambda by using target entropy since, ultimately, a low entropy on the target is a proxy for a confident classifier whose predictions are peaky. In other words, since \lambda regulates the effect of the correlation alignment, it also balances the capability of a classifier trained on the source to perform well on the target. Since in our pipeline \lambda is the only parameter related to domain adaptation, we deem our choice quite natural. In fact, other free parameters  (learning rate, early stopping) are not related to adaptation, but to the regular training of the deep neural network, which can be actually determined by using source data only - as we did in our experiments.
> --
>
> W 4. - If one leaves a subset of the target domain out and use its labels for validation, how different would the selected balancing cost \lambda differ from that by the entropy?
>
> The availability of a few labeled samples from the target domain would cast the problem into semi-supervised domain adaptation. Instead, our work faces the more challenging unsupervised scenario.
> Indeed, we propose an unsupervised method which lead to the same results of using labelled target samples for validation. This is shown in the top-right of Figure 2: the blue curve accounts for the best target performance, which is computed by means of target test labels - thus not accessible during training. Differently, the red curve can be computed at training time since the entropy criterion is fully unsupervised.
> Figure 2 shows that the proposed criterion is effectively able to select the \lambda which corresponds to the best target performance that one could achieve if one was allowed to use target label. Notice that the same does not happen for Deep CORAL (bottom-right) - and the reported results for that competitor were done by direct validation on the target.
> --

---

> > ### Author Response · Authors · 2017-12-28
> > **Response to AnonReviewer1 - part 2**
> >
> > W 5. - The cross-modality setup (from RGB to depth) is often not considered as domain adaptation. It would be better to replace it by another benchmark dataset. The Office-31 dataset is still a good benchmark to compare different methods and for the study in Section 5.1, though it is not necessary to reach state-of-the-art results on this dataset because, as the authors noted, it is almost saturated.
> >
> > In domain adaptation, the equivalence between domain and dataset is not automatic and some works have been operating in the direction of discovering domains as a subpart of a dataset (e.g., Gong et al. Reshaping Visual Datasets for Domain Adaptation - NIPS 2013). In this respect, the NYU dataset can be used to quantify adaptation across different sensor modalities within the same dataset.
> > The NYU experiment we carried out was also considered in the following recent domain adaptation works: Tzeng et al. “Adversarial Discriminative Domain Adaptation ICCV 2017” and Volpi et al. “Adversarial Feature Augmentation for Unsupervised Domain Adaptation” ArXiv 2017. We believe such experiment adds a considerable value to our work and we would like to maintain it.
> > In any case, after the reviewer’s suggestion, we are now running the Office-31 experiments. Preliminary results on the Amazon->Webcam split are in line with those already in the paper and coherent with the ones published in Sun & Saenko, 2016: Baseline (no adapt) 58.1%, Deep-Coral +5.9%, MECA +8.7% (Note that we use a VGG as a baseline architecture, while Sun & Saenko, 2016 use AlexNet).
> > ---
> >
> > Q 1. - I am not sure how the gradients were computed after the eigendecomposition in equation (8).
> >
> > As a common practice, we let the software library to automatically compute the gradients along the computation graph, given the fact that the additive regularizer that we wrote is nothing but a differentiable composition of elementary functions such as logarithms and square exponentiation. Although it’s possible to explicitly write down gradients with formulas, such explicit formalism is not of particular interest and we decided to remove such calculations from the paper in order to reduce verbosity.

---

> ### Comment · AnonReviewer1 · 2018-01-17
> **Comments after reading the rebuttal**
>
> The rebuttal addresses most of my questions. Here are two more cents.
>
> The theorem still does not favor the correlation alignment over the geodesic alignment. What Figure 2 shows is an empirical observation but the theorem itself does not lead to the result.
>
> I still do not think the cross-modality setup is appropriate for studying domain adaptation. That would result in disparate supports to the distributions of the two domains. In general, it is hard to adapt between two such "domains" though the additional pairwise relation between the data points of the two "domains" could help. Moreover, there has been a rich literature on multi-modality data. It is not a good idea to term it with a new name and meanwhile ignore the existing works on multi-modalities.

---

### Official Review · AnonReviewer2 · 2017-11-27
**New correlation alignment based domain adaptation method which results in minimal target entropy**

**Rating:** 7
**Confidence:** 5

**Review:**

Summary:
This paper proposes minimal-entropy correlation alignment, an unsupervised domain adaptation algorithm which links together two prior class of methods: entropy minimization and correlation alignment. Interesting new idea. Make a simple change in the distance function and now can perform adaptation which aligns with minimal entropy on target domain and thus can allow for removal of hyperparameter (or automatic validation of correct one).

Strengths
-  The paper is clearly written and effectively makes a simple claim that geodesic distance minimization is better aligned to final performance than euclidean distance minimization between source and target.
- Figures 1 and 2 (right side) are particularly useful for fast understanding of the concept and main result.


Questions/Concerns:
- Can entropy minimization on target be used with other methods for DA param tuning? Does it require that the model was trained to minimize the geodesic correlation distance between source and target?
- It would be helpful to have a longer discussion on the connection with Geodesic flow kernel [1] and other unsupervised manifold based alignment methods [2]. Is this proposed approach an extension of this prior work to the case of non-fixed representations in the same way that Deep CORAL generalized CORAL?
- Why does performance suffer compared to TRIPLE on the SYN->SVHN task? Is there some benefit to the TRIPLE method which may be combined with the MECA approach?


[1] Boqing Gong, Yuan Shi, Fei Sha, and Kristen Grauman. Geodesic flow kernel for unsupervised domain adaptation. In CVPR, 2012.

[2] Raghuraman Gopalan and Ruonan Li. Domain adaptation for object recognition: An unsupervised approach. In ICCV, 2011.

---

> ### Author Response · Authors · 2017-12-28
> **Response to AnonReviewer2**
>
> We are thankful for the provided comments and we will respond (A) to each query (Q) in detail.
>
>
> Q 1 - (a) Can entropy minimization on target be used with other methods for DA param tuning?  (b) Does it require that the model was trained to minimize the geodesic correlation distance between source and target?
>
> A 1 - (a) Let us point out that we are not minimizing entropy on the target as a regularizing training loss, as previous works did (Tzeng et al. 2015, Haeusser et al. 2017 or Carlucci et al. 2017). For the latter methods, entropy cannot be used as a criterion for parameter tuning, since it is one of the quantities explicitly optimized in the problem. Differently, we obtain the minimum of the entropy as a consequence of an optimal correlation alignment. Such criterion could possibly be used for other methods aiming at source-target distribution alignment. (b) Alignment does not *explicitly* require a geodesic distance. However, since the former must be optimal, it cannot be attained with an Euclidean distance, which is the reason why we propose the log-Euclidean one.
>
>
> Q 2. - It would be helpful to have a longer discussion on the connection with Geodesic flow kernel [1] and other unsupervised manifold based alignment methods [2]. Is this proposed approach an extension of this prior work to the case of non-fixed representations in the same way that Deep CORAL generalized CORAL?
> [1] Boqing Gong, Yuan Shi, Fei Sha, and Kristen Grauman. Geodesic flow kernel for unsupervised domain adaptation. In CVPR, 2012.
> [2] Raghuraman Gopalan,, Ruonan Li  and Rama Chellappa. Domain adaptation for object recognition: An unsupervised approach. In ICCV, 2011.
>
> A -2 The works [1,2] are kernelized approaches which, by either using Principal Components Analysis [1] or Partial Least Squares [2], a sequence of intermediate embeddings is generated as a smooth transition from the source to the target domain. In [1], such sequence is implicitly computed by means of a kernel function which is subsequently used for classification. In [2], after the source data are projected on hand-crafted intermediate subspaces, classification is performed.
> In [1] and [2], the necessity for engineering intermediate embeddings is motivated by the need for adapting the fixed input representation so that the domain shift can be solved. As a way to do it, [1] and [2] follow the geodesics on the data manifold.
> In a very same way, our proposed approach, MECA, follows the geodesics on the manifold (of second order statistics), but, differently, this step is finalized to better guide the feature learning stage.
> For all these reasons, MECA and [1,2] can be seen as different manners of exploiting geodesic alignment for the sake of domain adaptation.
>
>
> Q 3. - Why does performance suffer compared to TRIPLE on the SYN->SVHN task? Is there some benefit to the TRIPLE method which may be combined with the MECA approach?
>
> A 3 - As we argued in the paper, the performance on SYN to SVHN task is due to the the visual similarity between source and target domain whose relative data distributions are already quite aligned. Also note that TRIPLE already performs better than direct training on the target domain. This could be interpreted as a cue for TRIPLE to perform implicit data augmentation on the source synthetic data (and, indeed, the same could be done in MECA, trying to boost its performance by means of data augmentation). However, when more realistic datasets are used as source, such procedure becomes more difficult to be accomplished and that’s why, on all the other benchmarks, TRIPLE is inferior to MECA in terms of performance.

---

### Official Review · AnonReviewer3 · 2017-11-30
**This paper proposes a principled connection between correlation alignment and entropy minimization to achieve a more robust domain adaptation. The authors show the connection between the two approaches within a unified framework. The experimental results support the claims in the paper, and show the benefits over state-of-the-art methods such as DeepCoral.**

**Rating:** 8
**Confidence:** 4

**Review:**

The authors propose a novel deep learning approach which leverages on our finding that entropy minimization
is induced by the optimal alignment of second order statistics between source and target domains. Instead of relying on Euclidean distances when performing the alignment, the authors use geodesic distances which preserve the geometry of the manifolds. Among others, the authors also propose a handy way to cross-validate the model parameters on target data using the entropy criterion. The experimental validation is performed on benchmark datasets for image classification. Comparisons with the state-of-the-art approaches show that the proposed marginally improves the results. The paper is well written and easy to understand.

As a main difference from DeepCORAL method, this approach relies on the use of geodesic distances when doing the alignment of the distribution statistics, which turns out to be beneficial for improving the network performance on the target tasks. While I don't see this as substantial contribution to the field, I think that using the notion of geodesic distance in this context is novel.  The experiments show the benefit over the Euclidean distance when applied to the datasets used in the paper.

A lot of emphasis in the paper is put on the methodology part. The experiments could have been done more extensively, by also providing some visual examples of the aligned distributions and image features. This would allow the readers to further understand why the proposed alignment approach performs better than e.g. Deep Coral.

---

> ### Author Response · Authors · 2017-12-28
> **Response to AnonReviewer3**
>
> We are thankful for the detailed reading and careful evaluation of our work.
>
>
> By following the proposed suggestion, we added to the Appendix some t-SNE visualizations in which we compare our baseline network with no adaptation against Deep CORAL and MECA on the SVHN to MNIST benchmark. As the we observed, Deep CORAL and MECA achieve a better separation among classes - confirming the quantitative results of Table 1.
>
> Moreover, when looking at the degree of confusion between source and target domain achieved within each digit’s class, we can qualitatively show that MECA is better in “shuffling” source and target data than Deep CORAL, in which the two are close but much more separated. This can be read as an additional, qualitative evidence of the superiority of the proposed geodesic over the Euclidean alignment.
>
> These considerations and further remarks have been discussed in the revised paper (appendix).

---

### Decision · Program_Chairs · 2018-01-29
**ICLR 2018 Conference Acceptance Decision**

**Decision:**

Accept (Poster)

**Comment:**

This paper presents a nice approach to domain adaptation that improves empirically upon previous work, while also simplifying tuning and learning.